# Intravenous Administration of Human-Derived Mesenchymal Stem Cell-Conditioned Medium for Patients with General Malaise

**DOI:** 10.3390/jcm14165884

**Published:** 2025-08-20

**Authors:** Norihito Inami

**Affiliations:** Seihoku Clinic, 775 Takawa, Oshibedani, Nishi-ku, Kobe 651-2204, Japan; inami@shhc.jp; Tel.: +8178-998-1101

**Keywords:** conditioned medium, malaise, mesenchymal stem cell, quality of life

## Abstract

**Objectives:** Animal studies have demonstrated that mesenchymal stem cell-conditioned medium (MSC-CM) possesses various therapeutic effects, including anti-inflammatory and anti-fibrotic properties. This study investigated the efficacy and safety of administering MSC-CM as a treatment for patients with generalized fatigue. **Methods:** The MSC-CM used in this study was derived from human adipose tissue and umbilical cord-derived mesenchymal stem cells cultured in a medium free of animal-derived components to avoid the risk of infection. This MSC-CM has recently been shown to possess anti-inflammatory effects and has been reported to be safe for human administration. With the expectation of alleviating fatigue symptoms through its anti-inflammatory effects, it was administered to patients intravenously and by inhalation. Safety and changes in subjective symptoms were evaluated, and blood biomarkers related to inflammation and oxidative stress were measured. **Results:** In this trial involving 19 patients experiencing fatigue, no serious side effects were observed following MSC-CM administration. Nearly half of the patients reported symptom improvement after a single dose, and some exhibited signs of reduced inflammation. **Conclusions:** This report presents the first investigation of systemic MSC-CM treatment for generalized fatigue, paving the way for more targeted studies on dosage and treatment frequency. These findings offer new hope and possibilities for treating fatigue, providing valuable insights into the clinical application of MSC-CM.

## 1. Introduction

General malaise refers to physical and mental fatigue throughout the body, which impact an individual’s quality of life (QOL). Underlying causes of general malaise include conditions such as cardiovascular diseases, diabetes, and liver cirrhosis; therefore, it should not be underestimated [1,2,3,4]. Recently, up to 15% of survivors of COVID-19 infection experience long-term health effects, such as fatigue and cognitive decline, known as post-COVID-19 symptoms, or long COVID [5], highlighting the need for treatment to maintain QOL.

However, no appropriate treatments exist for general malaise beyond addressing the underlying condition, and only symptomatic treatments are available. For example, in patients with chronic fatigue syndrome, low 25-hydroxyvitamin D levels are common, and high-dose oral vitamin D3 therapy has been proposed. However, RCTs have shown that high-dose oral vitamin D3 does not improve the level of fatigue in patients with chronic fatigue syndrome [6]. Recently, increased inflammation and oxidative stress have been reported as potential causes of fatigue [7,8,9]. For example, it is known that elevated blood levels of CRP, an inflammatory marker, are observed in patients with chronic fatigue [10]. Abnormalities in the reactive oxygen species (ROS) clearance pathway have also been reported in patients with chronic fatigue syndrome and long COVID [11]. These reports suggest that the administration of drugs with anti-inflammatory and antioxidant properties, or drugs that regulate the internal environment, may be useful in improving fatigue.

Previous studies have shown that mesenchymal stem cells (MSCs) possess anti-inflammatory, anti-fibrotic, and pro-angiogenic effects, making them promising candidates for regenerative medicine [12]. Clinical trials have evaluated the use of MSCs in neurological diseases such as amyotrophic lateral sclerosis, demonstrating their efficacy in delaying disease progression [13,14]. Many of the therapeutic effects of MSCs result from the factors they release. Thus, it has become evident that acellular conditioned medium (MSC-conditioned medium [MSC-CM]), which contains factors released by MSCs during culture, also exhibits therapeutic effects [15]. MSC-CM is already known to possess anti-inflammatory [16,17] and antioxidant effects [18,19]. Actually, MSC-CM contains anti-inflammatory molecules such as IL10, TGFb, and PGE2 [20], as well as molecules that exhibit antioxidant properties, such as SOD2 and PRDX3 [21]. MSC-CM administration has been shown to attenuate lung injury in a mouse model of COVID-19 spike protein-induced lung injury [22].

By suppressing inflammation and oxidation, MSC-CM is expected to be effective in treating fatigue. This study was conducted to verify the safety and therapeutic efficacy of MSC-CM as a novel therapeutic agent for patients with general fatigue. MSC-CM was administered to patients with general fatigue that had not improved with standard treatment. The focus of this study was on systemic inflammation and oxidative stress, which are considered to be factors in fatigue expression. To ensure patient safety, MSCs were cultured in a medium free of animal-derived components to reduce risk of infection. The selected MSC-CM have passed viral and animal safety tests, providing a high level of assurance for patients undergoing treatment.

MSC-CM has been used in several human clinical trials, including those investigating chronic pain treatment [23]; however, these studies employed small, locally administered doses. To the best of my knowledge, this is the first study to confirm the safety of systemic MSC-CM administration in patients with general fatigue unresponsive to other treatments, making this a novel and noteworthy research finding in this field.

## 2. Materials and Methods

### 2.1. Ethical Considerations

This study was conducted in accordance with the Ethical Guidelines for Life Science and Medical Research Involving Human Subjects (issued on 23 March 2021 [partially revised on 10 March 2022 and 27 March 2023]). Ethical approval was obtained from the Seihoku Clinic ethical review committee (approval numbers 22000152 and 20220819-2), and this study was registered in the Japan Registry of Clinical Trials (jRCT1051220089). Written informed consent for treatment was obtained by the attending physician from the research participants.

### 2.2. Participants

Data were collected from 19 patients with general fatigue (10 men and 9 women, aged 37 to 91 years) who received three intravenous (IV) doses of MSC-CM or 12 inhaled (INH) doses at Seihoku Clinic between October 2022 and August 2023. Patient backgrounds are listed in Appendix A. INH administration of MSC-CM was performed at the patient’s request (No.19).

### 2.3. Research Participant Selection Policy

1.Patients who had not achieved satisfactory treatment effects with standard treatments provided by a general physician.2.Patients who opted not to undergo standard drug treatments, owing to concerns about side effects, and whose physician deemed treatment with the MSC-CM to be optimal.

The patient selection criteria were as follows:Those aged 18 years or older;Those with normal consent capacity;Those who provided written informed consent;Those for whom the physician recognized a need for treatment.

Exclusion criteria

Patients who met any of the following conditions were excluded:1.Those with a history or suspicion of dementia;2.Those who used drugs and stimulants;3.Those who were pregnant or breastfeeding;4.Those who were judged as unsuitable by the attending physician.

### 2.4. Types of MSC-CM Used for Treatment

Treatments were conducted using human adipose tissue-derived MSC-conditioned medium (AD-CM) or human umbilical cord-derived MSC-conditioned medium (UC-CM). The MSC-CM preparation protocol was previously described [23,24].

In briefly, adipose and umbilical cord tissue samples for AD-CM and UC-CM production were obtained respectively from two Japanese women in their 20s who passed a virus-negative test and who provided a consent form. AD MSC and UC MSC have been grown using an animal origin-free (AOF) medium (sf-DOT; BIOMIMETICS SYMPATHIES Inc., Tokyo, Japan) as reported previously [24]. AD MSC and UC MSC at passages 3–5 were cultured to reach approximately 80% confluent and were incubated for 3 days in the AOF medium. The CM was then harvested and centrifuged at 3000× *g* for 5 min to remove cell debris, and then the CM was collected and passed through a 0.22 μm filter to eliminate potential pathogens. The production of MSC-CM was carried out in cell processing center (CPC). The total protein amounts of AD- and UC-CM were measured at 12.8 μg/mL and 13.6 μg/mL, respectively. The amounts of hepatocyte growth factor (HGF) and exosomes in these MSC-CM were reported previously [24].

### 2.5. MSC-CM Administration Method

A mixture of 10 mL of human AD-CM and 500 mL of physiological saline (Otsuka Pharmaceutical Co., Ltd., Tokyo, Japan) was infused into the median cubital vein over approximately 24 h. If no improvement in subjective symptoms was observed after administration of AD-CM, UC-CM was used for the next administration (five patients, Nos. 003, 004, 010, 012, and 018).

INH administration was performed for a single long-Covid patient (No. 019) by mixing 1 mL of AD-CM with 4 mL of physiological saline, which was then administered using a nebulizer (Shenzhen IMDK Medical Technology Co. Ltd., Shenzhen, China).

Malaise was assessed during interviews using an 11-point numerical rating scale (NRS), where 0 indicated “no malaise” and 10 indicated “the most intense malaise imaginable.” The following items were evaluated: (1) current malaise, (2) typical malaise in the past 24 h, and (3) most intense malaise in the past 24 h. Additionally, the impacts of malaise on the QOL indicators were evaluated using an 11-point NRS, with 0 as “no problems” and 10 as “completely a problem.” The evaluated items included the following: (1) general activities of daily life, (2) mood and emotion, (3) walking ability, (4) regular work (including work outside the home and daily chores), and (5) interpersonal relationships. An open-ended interview was also conducted during initial assessment regarding awareness of chief complaints and other symptoms.

### 2.6. Administration Schedule

The schedules for treatment administration, blood collection, and other details are presented in Figure 1.

Patients were interviewed a week prior to the initial treatment administration, and the CM was administered via the IV route three times monthly and via the INH route 12 times weekly.

### 2.7. Blood Collection

Serum was collected before treatment and 1 month after the third administration. The levels of adiponectin, C-reactive protein (CRP), and serum total antioxidant status (STAS) were measured. Adiponectin and CRP measurements were outsourced to BML, Inc. (Tokyo, Japan). STAS was assessed using a Redox Assay Total Antioxidant Capacity Measurement Kit (Metallogenics Inc., Chiba, Japan) and expressed as ascorbic acid equivalent.

### 2.8. Statistical Analyses

The levels of all parameters assessed before and after three administrations of MSC-CM were subjected to the Kolmogorov–Smirnov test to assess normality. Considering that the data did not follow a normal distribution, a two-tailed Wilcoxon signed-rank test was performed on all parameters. The sample size was 19, and a significant difference level of 0.05 was established for the *p*-value cut-off. All statistical analyses were performed using R (version 4.3.1).

## 3. Results

### Changes in Subjective Symptoms

The medical backgrounds of the 19 patients analyzed (10 males, 9 females; mean age, 73.2 ± 14.0 years, from 37 years to 91 years) are summarized in Appendix A. Among them, 14 had cardiovascular disease, 5 had diabetes, 6 experienced pain, and 2 had respiratory disease. Additionally, 5 patients had a history of cancer. The total exceeds the number of participants because some patients had multiple diseases simultaneously.

The results of the patients’ subjective symptoms and blood collection are presented in Appendix A, with changes in subjective symptoms shown in Table 1. Improvements in chief complaints were observed in 10 of 19 patients following initial treatment (Table 1). Patient No. 19 presented with aftereffects of coronavirus infection and reported improvement in general fatigue after INH administration of CM. However, multiple administrations did not lead to an increase in the number of patients reporting improvement in their chief complaint. Conversely, the number of patients who exhibited improvements in aspects other than their chief complaint increased with multiple administrations. Some of these responses included improved hypertension (1 patient), decreased number of headaches (1 patient), improved bowel movements (3 patients), appetite (1 patient), sputum (1 patient), and better sleep (6 patients). No adverse events were observed during all CM administration.

The inflammatory marker CRP has been reported to be associated with fatigue levels in older individuals [25]. Additionally, oxidative stress has been identified as an important factor in idiopathic chronic fatigue [26]. Adiponectin is expressed at low levels in various conditions, such as coronary artery disease [27] and diabetes [28], and can be a background factor for general malaise. Therefore, maintaining high adiponectin levels may be useful in suppressing fatigue. In this study, serum levels of CRP, STAS, and adiponectin were analyzed before and after MSC-CM.

Changes in mean CRP levels pre- and post-MSC-CM administration were not significant (0.127 vs. 0.069, *p* = 0.063, Figure 2a); however, a decreasing trend was observed (Figure 2a). Notably, in three patients (Nos. 006, 008, and 016) whose CRP levels exceeded the standard threshold of 0.3 mg/dL pre-MSC-CM administration, levels decreased to within the standard range afterward (Figure 2b). No changes in STAS were observed (0.530 vs. 0.521, Appendix A). Adiponectin levels tended to increase following three MSC-CM administrations; however, this change was not significant (13.33 vs. 15.18, *p* = 0.071, Appendix A).

Interview results regarding malaise, specifically typical malaise in the past 24 h, showed a significant decreasing trend (6.11 vs. 4.11, *p* = 0.007, Figure 3). Furthermore, interview results on the impact of malaise on QOL indicators, such as general activities of daily life, mood and emotion, walking ability, and interpersonal relationships, showed a decreasing trend without significance (Appendix A). In particular, the impairment degree of daily work showed a significant decrease (4.11 vs. 2.11, *p* = 0.020, Figure 4)

In this study, MSC-CM were administered to 19 general fatigue patients. Approximately half of the patients reported a reduction in fatigue. No adverse events were reported in any of the treatments. Blood tests showed that CRP levels decreased in three patients with particularly high CRP levels after MSC-CM administration. This finding suggests that the anti-inflammatory effects of MSC-CM may be effective in reducing general fatigue. Further research is expected to explore the optimization of MSC-CM administration for the treatment of fatigue.

## 4. Discussion

The potential of MSC-CM administration to improve symptoms in patients with general fatigue was analyzed in this study. A safety evaluation and a survey of changes in subjective symptoms following MSC-CM administration were conducted. MSC-CM was administered via IV (18 patients) and INH (1 patient) routes to patients whose malaise did not resolve with conventional treatment. During these administrations, no infections were observed, and CRP levels, an inflammation marker, remained below the standard value in all patients. No other side effects were observed (Appendix A). Therefore, in this study, the IV and INH methods used to administer AD-CM and UC-CM were safe; however, observations with a larger sample size are warranted in the future.

Improvements in subjective symptoms were observed in 10 of the 19 patients after a single administration. To the best of my knowledge, this is the first study to demonstrate that systemic MSC-CM administration could improve malaise. While this study focused on the intravenous administration of 10 mL of MSC-CM, optimization of the dose and frequency of administration is necessary to improve efficacy further. In one patient, general fatigue due to the aftereffects of COVID-19 infection was improved by INH administration of MSC-CM (No. 19 in Appendix A). Because post-COVID-19 patients often experience general fatigue accompanied by shortness of breath, INH administration may be more effective in improving fatigue due to respiratory failure. Further study with a larger number of patients is needed to verify this possibility. Future studies should explore additional administration methods, such as pulmonary artery administration, which may prolong cytokine circulation in MSC-CM, deliver more cytokines, and increase the amount of MSC-CM.

Regarding patients’ subjective symptoms in the present study, approximately half of the patients showed improvements in their chief complaints following the initial treatment administration, but no additional benefit was observed with multiple administrations (Table 1). This suggests that frequent administration may not be very effective; however, an effect on chief complaints was observed in terms of subjective symptoms in half of the patients for whom no satisfactory treatment effect was observed with standard treatment. Future studies should increase the sample size, compare single administration and frequent administration, and clarify the efficacy rate.

Many of the patients had conditions commonly recognized as background factors for fatigue, including 14 with cardiovascular disease, 5 with diabetes, 6 with pain, 2 with respiratory disease, and 5 with a history of cancer (Appendix A). In the current study, the correlation between patient background and treatment outcomes could not be clarified; however, it will be necessary to include many patients with similar backgrounds in the future to clarify the possibility of a correlation. In addition, improvements were observed, including improvement in high blood pressure, and reduced headache frequency, and improvement in bowel movements, appetite, phlegm, and sleep. Further case accumulation is needed to clarify these secondary treatment effects.

While no clear standard values for CRP based on solid evidence exist, the University of Tokyo Division for Health Service Promotion established a standard range of 0–0.3 mg/dL, which was adopted in this study. Three patients (Nos. 006, 008, and 016) had CRP levels above the standard value before treatment administration. These three patients’ CRP levels returned to baseline after three AD-CM infusions (Figure 2b). MSC-CM contains a high concentration of anti-inflammatory molecules, such as IL10, TGFβ1, and PGE2 [20], suggesting that AD-CM administration may suppress inflammation in the body. Additionally, Nos. 006 and 008 had cardiovascular diseases (Appendix A). In previous animal experiments, MSC-CM and the MSC exosomes improved cardiovascular diseases [29,30], suggesting that the present treatment may have improved the underlying condition. However, as no tests for cardiovascular diseases were conducted, the mechanism remains unclear. In the future, the treatment effects of cardiovascular diseases should be examined based on patient background, increased sample size, and administration amount and frequency.

To address the limitations of this study, several factors must be considered. The lack of a control group means that we cannot rule out placebo effects or other confounding factors that may influence the observed improvements. Future research should include randomized controlled trials to strengthen the evidence supporting the therapeutic effects of MSC-CM. Finally, although improvements in subjective symptoms were reported, objective biomarkers for assessing fatigue and lethargy were not adequately evaluated. Incorporating objective assessments such as inflammatory cytokine levels (for example, TNF-α), inflammatory markers other than CRP, and comprehensive quality of life (QOL) measures would provide a more robust understanding of the efficacy of fatigue treatment. Although many challenges remain, the safety and efficacy of MSC-CM for the treatment of general fatigue are promising, and further research is expected to lead to the development of more effective MSC-CM therapy.

A key issue in developing MSC-CM into pharmaceuticals in the future is assessing the equivalence of MSC-CM quality. MSC-CMs are complex products containing various secreted proteins, exosomes, and small molecules such as PGE2 at various concentrations. To demonstrate the equivalence of MSC-CM for treatment, active ingredients (anti-inflammatory ingredients) for the treatment of fatigue must be identified, and the concentration of this effective factor must be used as an evaluation item in the MSC-CM standard. While there are many challenges in drug development, further research into the clinical application of MSC-CM is warranted, as it is expected to lead to the development of treatments based on new mechanisms of fatigue.

## 5. Conclusions

Systemic fatigue significantly reduces quality of life, but no effective treatment has been found. In this study, systemic administration of MSC-CM, which has anti-inflammatory properties, improved fatigue symptoms in approximately half of the patients. Although further investigation into the dosage and frequency of administration is necessary, MSC-CM administration is considered to be an effective candidate for the treatment of fatigue.

## Figures and Tables

**Figure 1 jcm-14-05884-f001:**
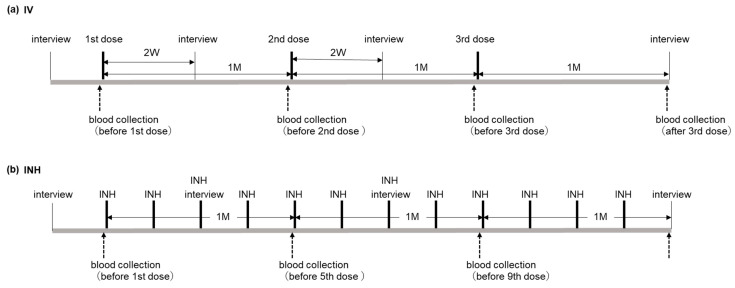
Administration schedule. (**a**) Schedule of intravenous (IV) administration to research participants. Patient interviews were conducted approximately 1 week before initial administration, with the conditioned medium administered thrice every month. Blood was collected immediately before each administration and 1 month after the third administration, and the serum was stored. Interviews regarding changes in subjective symptoms were conducted 2 weeks after the first and second administration and 1 month after the third administration. (**b**) Schedule of inhalation administration (INH) for research participants. Patient interviews were conducted approximately 1 week before initial administration, with 12 administrations given every week. Blood was collected immediately before the initial, fifth, and ninth administrations and 1 week after the 12th administration, and the serum was stored. Interviews were conducted before the third and seventh administration and 1 week after the 12th administration. Abbreviations: IV, intravenous; INH, inhalation.

**Figure 2 jcm-14-05884-f002:**
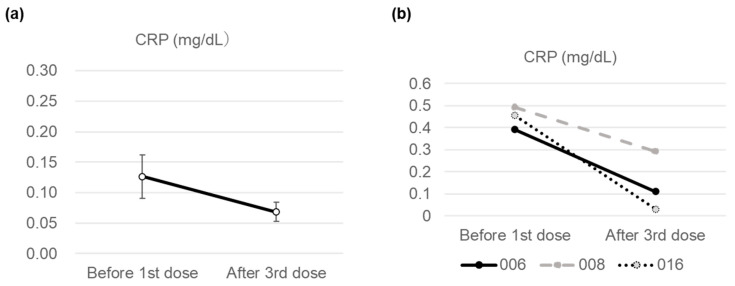
Changes in blood collection values: serum factors before the administration of the conditioned medium and after the third administration. Mean C-reactive protein (CRP) level across 19 patients (**a**) and changes in participants whose CRP level exceeded the standard value (0.3 mg/dL) before administration (**b**). Error bars are the standard errors. Abbreviations: CRP, C-reactive protein.

**Figure 3 jcm-14-05884-f003:**
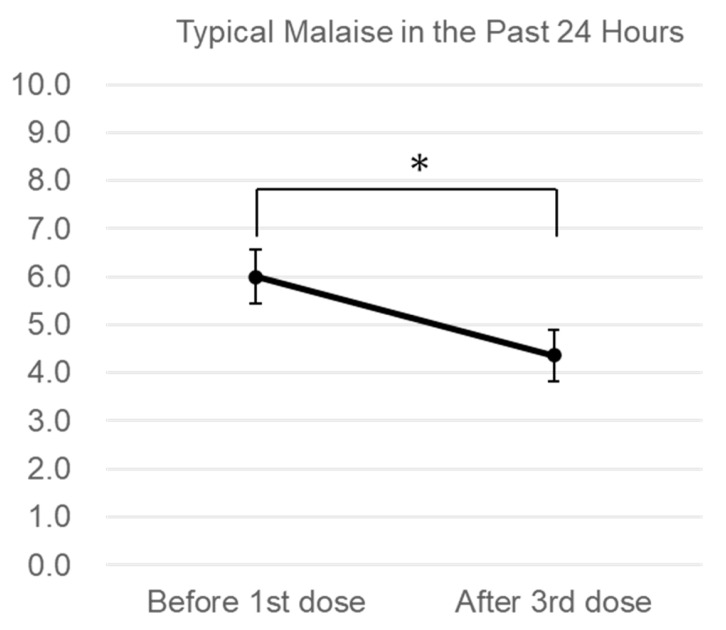
Changes in general malaise. The malaise was reported by patients during the interviews before the administration of the conditioned medium and after the third administration (after the 12th administration in the case of inhalation) on an 11-point scale from 0–10, with 0 being “no malaise,” for typical malaise in the past 24 h. * *p* < 0.05. Error bars are the standard errors.

**Figure 4 jcm-14-05884-f004:**
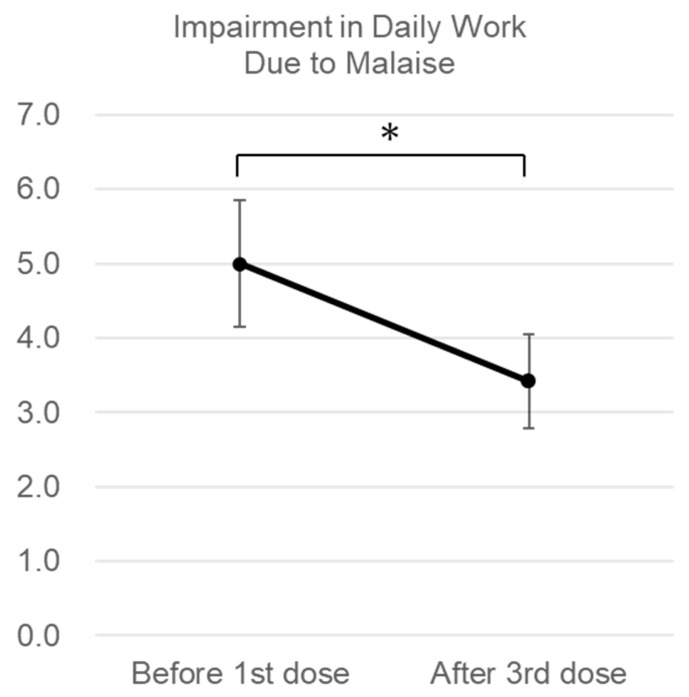
Changes in the impact of malaise on daily activity. In interviews with subjects before and after the third dose of conditioned medium (after the 12th dose for inhalation), subjects rated the degree to which fatigue interfered with their general activities of daily work on an 11-point scale from 0 to 10 (0 being “no interference at all”). * *p* < 0.05. Error bars are the standard errors.

**Table 1 jcm-14-05884-t001:** Changes in self-reported subjective symptoms. Percentage of patients with improvements in chief complaints, other symptoms, and no improvement are shown regarding the subjective symptoms of participants who underwent interviews after the third administration.

	Improvement of Chief Complaints	Improvement in Other Symptoms	No Improvement
1st dose	52.6% (10/19)	15.8% (3/19)	31.6% (6/19)
2nd dose	52.6% (10/19)	21.1% (4/19)	26.3% (5/19)
3rd dose	52.6% (10/19)	26.3% (5/19)	21.1% (4/19)

## Data Availability

The datasets generated during and/or analyzed during the current study are available from the author on reasonable request. However, disclosure cannot be provided for information that may lead to the identification of individual research participants in accordance with the personal information protection law.

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
