# Peer review of "Intravenous Administration of Human-Derived Mesenchymal Stem Cell-Conditioned Medium for Patients with General Malaise"

_jcm, 2025, doi:10.3390/jcm14165884_

Round 1
Reviewer 1 Report
Comments and Suggestions for Authors
The article titled "Intravenous administration of human-derived mesenchymal stem cell-conditioned medium for patients with general malaise," authored by Norihito Inami, reports for the first time that MSC-CM administration in patients with general fatigue is unresponsive to other treatments and improves inflammation-related fatigue.
This article will highlight a new and significant research discovery in the field.
However, I have some important suggestions: measuring cytokines from blood samples taken before and after treatment using ELISA or similar methods could enhance the manuscript.
Author Response
Reply to reviewer 1
Thank you for your suggestion.
I agree that measuring cytokines from blood samples would certainly improve the quality of the paper. Unfortunately, however, we no longer have any serum samples left after using them for measuring CRP and STAS. I believe that future studies should analyze cytokine concentrations before and after treatment. I have added this information to the Discussion section of the manuscript (page 14, lines 12-15).
Reviewer 2 Report
Comments and Suggestions for Authors
This is simple clinical study on treating fatigue symptoms by using MSC-CM Intravenously. Although there lacks concrete scientific evidence in mechanism, about half of the patients gained some improvement in fatigue. Since many reports show that exosomes in MSC-CM exert multiple bio-activities to improve inflammatory conditions, both in animal experiment and human trials, the result of this clinical study is convincing. However, some important matter is Ignored in the manuscript. First, the author should provide details of the components in the MCS-CM, such as a total protein concentration and dose of exosomes. Secondly, the author should give more specific evidence to show the improvement in fatigue, such as the improvement in tonicity of muscles,grip force and other functions. Third, clearly, the complexity of the components in MSC-CM and the difficulty in quality control hamper its becoming a commercial drug. Thus, the author should give some estimation in Discussion about the possibility in future to expand the clinical uses of MSC-CM in fatigue and other chronic general malaise. Besides, the problems it will meet should also be discussed.
Author Response
Reply to reviewer 2
Thank you very much for your valuable comments.
The total protein amounts of AD- and UC-CM were measured at 12.8 mg/ml and 13.6 mg/ml, respectively. In reference No. 24 paper (PLOS One 2025, 20, e0322497), the amounts of HGF and exosomes used in this study were quantified previously. I believe these values should be established as reference values for MSC-CM when it is developed into a pharmaceutical product in the future. This content has been added on page 3, lines 38-40 of the manuscript.
Finally, I have described the challenges that arise when developing MSC-CM as a pharmaceutical product on page 14, lines 19-27 of the manuscript.

Reviewer 3 Report
Comments and Suggestions for Authors
1. [Abstract] "Fatigue is a symptom associated with various diseases. When treatment is ineffective and symptoms become chronic, it interferes with daily life." - these sentences are unacceptable, please start the manuscript with more attractive thoughts.
2. [Figures] Figures 1-4: All these figures could be presented in single collage; into the present form the value of each graph is not justified, because the biochemical values should support the self-reported subjective parameters. "Figure 4. Changes in the impact of malaise on daily activity" - all these five parts could be presented into the single graph; into the present form the value of each graph is not justified. 3. [Figure] "Scheme 1. Administration schedule" - this is not the scheme, but separata figure. 4. [Tables] "Table 1. Patient background." and "Table 2. Changes due to conditioned medium administration. " - The tables contain only raw data without deep analysis; these raw datas should be transferred to the supplementary tables. 5. [Methods] Am I correct in understanding from the manuscript that the drug was manufactured in a laboratory without adhering to Good Manufacturing Practice (GMP) requirements? This point should be disclosed more clearly. 6. [Discussion] Using condition media derived from various cell types poses the risk of complications caused by the type of cell sources, cell culturing, cellular stress, and other ways for the senescent drift induction [https://pubmed.ncbi.nlm.nih.gov/38031201/ ] [https://pubmed.ncbi.nlm.nih.gov/36732079/ ]. Induced senescence in affected tissues could directly result from these interventions, which makes it difficult to diagnose patients with systemic fatigue. 7. Summary: The presented manuscript reports on an interesting clinical study of using a conditioned medium to treat patients for whom the use of such a high-risk ATMP-based drug could not be previously approved. Efforts like this, which expand clinicians' ability to use new drugs, should be supported. However, the analysis in this manuscript is extremely poor. The graphs merely visualize the data, and there is no in-depth analysis of the results or the potential of such studies. It seems as if the authors are disappointed in their work and are trying to publish it as quickly as possible. I do urge the authors to pay closer attention to in-depth analysis, especially when subjective data is confirmed by biochemical analysis. They should also discuss the safety and potential applications of these types of biological drugs. This point I also urge the authors to conduct public outreach for the entire TERMIS community.Author Response
Reply to reviewer 3
Thank you very much for your valuable feedback.
- Based on your feedback, I have changed the objectives section in the abstract.
- After rerunning the analysis, I found significant improvements in Figures 4a, 4b, and 5d. Therefore, I am presenting each item as a separate graph.
- As you pointed out, Scheme 1 was renamed to Figure 1.
- As you pointed out, Tables 1 and 2 have been renamed to Supplementary tables 1 and 2.
- The production of MSC-CM was carried out in a cell processing factory (CPC). This has been added to page 3, lines 37-38.
- As described in Section 2.4, the MSC-CM used in this study was extracted from cells that appear to be non-senescent, which promotes the proliferation of AD and UC MSCs from passages 3 to 5. Furthermore, as described in reference No. 24 (PLOS One 2025, 20, e0322497), it has been administered to 55 patients with various diseases, with no adverse events observed.
- In this study, patients with CRP levels above the reference range experienced an improvement in subjective fatigue assessment and a decrease in CRP levels after administration of MSC-CM. Although the objective therapeutic effect was poor, no adverse events were observed in the patients who received the treatment. This study is the first to demonstrate the safety and therapeutic effect of MSC-CM administration in patients with fatigue, and I think it is worthy of publication. The Introduction, Results, and Discussion sections have been significantly expanded to provide a detailed discussion of the significance and issue of this study. Additionally, a paragraph addressing challenges for future clinical studies and MSC-CM formulation development has been added to the Discussion section (page 14, lines 18-26).
Round 2
Reviewer 3 Report
Comments and Suggestions for Authors
The author has addressed all of my comments satisfactorily; however, the clarity of the data and figures presentation need to be improved.
Author Response
Reply to reviewer 3
Thank you very much for your suggestion.
Based on your suggestion, I have made significant changes to the figures in my paper. Specifically, I have focused on the parts of the figures that need to be emphasized and omitted other parts. I have tried to make the figures easier for readers to understand.